# Evaluating the Resonance of Official Islam in Oman, Jordan, and Morocco

Annelle Sheline

Middle East Program, The Quincy Institute, Washington, DC 20006, USA; annelle@quincyinst.org

**Abstract:** Acts of political violence carried out by Muslim individuals have generated international support for governments that espouse so-called "moderate Islam" as a means of preventing terrorism. Governments also face domestic skepticism about moderate Islam, especially if the alteration of official Islam is seen as resulting from external pressure. By evaluating the views of individuals that disseminate the state's preferred interpretation of Islam—members of the religious and educational bureaucracy—this research assesses the variation in the resonance of official Islam in three different Arab monarchies: Oman, Jordan, and Morocco. The evidence suggests that if official Islam is consistent with earlier content and directed internally as well as externally, it is likely to resonate. Resonance was highest in Oman, as religious messaging about toleration was both consistent over time and directed internally, and lowest in Jordan, where the content shifted and foreign content differed from domestic. In Morocco, messages about toleration were relatively consistent, although the state's emphasis on building a reputation for toleration somewhat undermined its domestic credibility. The findings have implications for understanding states' ability to shift their populations' views on religion, as well as providing greater nuance for interpreting the capacity of state-sponsored rhetoric to prevent violence.

**Keywords:** official Islam; resonance; Oman; Jordan; Morocco





## 1. Introduction

The 9/11 attacks prompted many Muslim-majority states to shift the content of state-sanctioned religious messaging, or "official Islam." Regime elites were motivated both by the desire to prevent—or to be seen as trying to prevent—acts of violent extremism, but also to be perceived as contributing to the counterterror security agenda pursued by powerful states like the US.

These shifts in the content of state-sanctioned Islam put employees of state-run religious bureaucracies in the position of having to promote an interpretation of Islam that may have differed significantly from their own spiritual beliefs. This article evaluates perceptions of official Islam, focusing in particular on the views of individuals responsible for disseminating it, namely, the employees of the Ministry of Religious Affairs and educators that operate under the auspices of the Ministry of Education. How do these individuals view the religious discourse they are charged with spreading?

In contexts where the state is a constitutionally sanctioned religious actor, it generates a form of centralized religious messaging; in the Arabic-speaking countries of the Middle East and North Africa (MENA) where Islam is the official religion, this is known in Arabic as "official Islam" (al-islam al-rasmi). Official Islam is primarily manufactured by state religious institutions like the Ministry of Religious Endowments/Affairs, as well as through other sources of official messaging like the Ministry of Education. Official Islam is disseminated through state-controlled mosques, state-affiliated religious institutions, religious educational content in public schools, as well as institutions of higher education for religious scholars ('ulama, sing. 'alim). My definition draws on that used by Robbins and Rubin: "official Islam refers to the elements of religious authority that are under

the direct or indirect control of the regime. while seeking to operationalize it for data collection purposes.[1]

Directly assessing a population's view on their government's action through public opinion polls presents a dilemma in non-democratic contexts, as exist throughout much of the Middle East and North Africa (MENA). Previously, much analysis of the interaction between Islam and politics tended to assume that official Islam lacked credibility based on the assumption that these non-democratic governments suffered from a general legitimacy deficit, which extended to their production of official Islam.[2] Instead the religious views of MENA societies are seen as more accurately reflected by Islamist groups. Islamists are here defined as any of a group of individuals and/or movements that advocate for expanding the role of Islam in public life. Much research has focused on Islamist movements; their electoral successes were often taken to mean that their positions corresponded to the views of the general public.[3] Yet as a result of the historical repression of alternative ideologies, political opposition tends to be dominated by Islamists, which can overinflate the apparent popularity of their stances, leading to perhaps an overemphasis on their views and actions.[4]

This consensus has started to shift, as more scholars have demonstrated that official Islam does not uniformly lack credibility: this article contributes to this growing body of work.[5] Throughout much of the MENA region, the state has heavily influenced Islam, a trend that has grown more pronounced as a result of fears of violent extremism, prompting greater securitization of Islam. Islamist groups operate in a religious arena of the state's making, therefore analysis that dismisses the state's religious messaging fails to account for a significant causal actor. Even an individual that rejects the official narrative was likely to have been educated in schools required to use state-sanctioned curricula, to have heard sermons in mosques subject to state surveillance and could not have avoided exposure to state-sponsored discourses: her (op)position is shaped by the state's articulation of Islam. This is not to say that she would not have access to alternative interpretations of Islam not sanctioned by the state, but to highlight the often-overlooked role of the state in expanding its reach over religious actors and institutions.[6]

This article seeks to evaluate the resonance of official Islam in three national contexts. While the concept of "resonance" is long established in sociological research, and most typically employed by political scientists to evaluate social movements, resonance here refers to the extent to which a given discourse is experienced as corresponding to what an individual recognizes as familiar and generally accepted within a given social setting.[7] Note that resonance does not necessarily correspond to an individual's beliefs, but instead reflects what the individual recognizes as a socially prevalent narrative. Regime elites seek to use ideas about national identity and religion established during the process of nation-building to frame official religious discourse in the contemporary context; when the contemporary frames resonate with those first established during the process of nation-building, official religious discourse is more likely to be reproduced rather than challenged.

The research operates on the understanding that an individual's existing religious identity is constructed from a combination of factors, including state religious education, familial traditions, individual spiritual experiences, and many other possible influences. This construction, which an individual may perceive as a deeply rooted component of their identity, is shaped partly by external influences and is subject to change. However, the notion of resonance posits that certain frames will be experienced by a given individual as corresponding to the existing ideas that they know are dominant in society. Individuals are more likely to accept discourses that correspond to these commonly held frames,

---

1   (Robbins and Rubin 2013).
2   See, e.g., (Dekmejian 1980; Keddie 1998; Kepel 2002; Mandaville 2007; Haklai 2009).
3   See, e.g., (Esposito 1984; Eickelman and Piscatori 1996; Wickham 2002; Masoud 2014).
4   (Masoud 2014; Schwedler 2012).
5   (Akbarzadeh and Saeed 2003; Henne 2013, 2017; Öztürk 2016; Öztürk and Sözeri 2018).
6   (Cesari 2014; Fabbe 2018; Feuer 2017; Wainscott 2017; Zubaida 1989).
7   (Benford and Snow 2000).

whereas discourses that do not correspond to that which is seen as common knowledge may be viewed as intended to manipulate and are therefore more likely to be resisted. In other words, when religious messaging resonates with what an individual recognizes as a nationally shared religious identity, she or he is more likely to be willing to reproduce the messaging, but if it does not resonate the individual will be more likely to challenge it.

The state can influence what is publicly known but cannot control all of it. In general, the process of establishing the parameters of what is accepted as historically true requires several generations of time and a hegemonic level of control that eludes most state structures. In addition, the successful imposition of top-down identities builds on existing identities.[8] As a result, even if a state theoretically began the process of establishing its preferred version of religious discourse at the earliest possible opportunity, i.e., following the introduction of the indoctrinating instruments of the modern state, specifically mass education, certain aspects of common knowledge always elude control.

Frame analysis offers insights for understanding how regime elites disseminate their desired discursive interpretations, although it tends to highlight the short-term effects of strategically choosing language to influence public opinion. Framing is loosely defined as "a political process by which actors, such as state elites, seek to impose their definition of a political reality."[9] Scholarship on framing explains how a given social actor uses a frame to encourage audiences to adopt a preferred viewpoint and act accordingly; however there has been relatively little study of the overlap between the processes of nation-building, which play out over the course of generations, and the processes of framing, which have more typically been analyzed for their short-term effects. Yet nation-building can be seen as the process of erecting the frames through which the population of a given nation-state understands its history and heritage, the shared points of cultural resonance that allow for a persistent sense of shared national identity. Culturally resonant frames are the result of the nation-building process.

## 2. Theoretical Framework

The extent to which both religious bureaucrats and non-bureaucrats are willing to reproduce the state's preferred articulation of religion, or "official Islam," is analyzed. I view this willingness as reflecting an individual's experience of resonance, which I define as the match between an individual's expressed views and the position established by official religious discourse. The concept of resonance has been most thoroughly developed by social movement theorists in the context of evaluating the effectiveness of a given frame to inspire social action. In their influential article on framing, Benford and Snow identified "four sets of factors concerned with the conditions that affect framing efforts," including "the cycle of protest in which the movement is embedded," a signal that their theorization of frame resonance is deeply embedded in their focus on social movements.[10] The concept of frame resonance offers a means of evaluating state-led discourses as well, although the specific dynamics at play when a frame is used by an authoritarian state, as opposed to a social movement, have been under-theorized.

Frame analysis uses the concept of "resonance" as a causal factor associated with frames that successfully inspire action. Credibility and salience constitute two key features of resonance. Credibility is seen as the result of frame consistency, empirical credibility, and the credibility of the frame articulators. Frame consistency refers to, "the congruency between a social movement organization's articulated beliefs, claims, and actions."[11] The consistency of a given frame with previously expressed beliefs, claims, or actions can be understood as corresponding to discourses established through the nation-building process. When an actor posits a frame that aligns with existing expectations, it is more likely to be seen as exhibiting consistency.

8   (Smith 2003).
9   (Jourde 2007; Goffman 1974).
10  (Benford and Snow 1988).
11  (Benford and Snow 2000).

Theorists have sought to overcome the conception of resonance as inherent to a given frame or cultural object, and instead think of it as an "emergent collective act" that reflects the importance of an individual's evaluation of other people's reactions to a given frame.[12] McDonnell et al. emphasize the interactional aspect of resonance, and their analysis is relevant to thinking about the ways in which individuals look to each other for signals of how and when to act, and who and what to believe.[13] In the context of either reproducing or challenging official religious discourse, interviewees' responses indicated the influence of others' opinions. Individuals in authoritarian contexts tend to be keenly aware of upholding the collective project of reifying the ruler's authority.[14]

This article's primary argument is that official religious discourse is more likely to resonate if it adheres to two criteria: consistency and internality. Consistency means adhering to shared understandings of religious identity; internality refers to directing religious messaging at a population seen as internal, i.e., co-religionists and/or fellow citizens.

Consistency: A shared understanding of religious identity is itself the result of a process of top-down messaging pertaining to a population's spiritual heritage and history. Scholars of nationalism have long analyzed the various means by which political peoples are constituted; however, religious identity is less typically viewed from a nationalist perspective and is more often noted for its capacity to transcend national boundaries than be limited by them.[15] Yet, in contexts where the state seeks to establish a monopoly over the legitimate expression of religious discourse, religion bears the marks of state influence and, as a result, takes on a national character.[16]

The importance of maintaining consistency with an existing understanding of religious identity—even one imposed through top-down processes—is not limited to a MENA context. Religions in general derive authority through their adherence to foundational texts, such that would-be change agents will be most successful if they can frame their desired alteration as reflecting scripture. Holy texts often contain incongruent behavioral prescriptions that necessitate interpretation, the veracity of which is judged on the basis of the interpreter's authority. Constructivism has come to dominate scholarly views of national identity; adopting a fully constructivist approach to religious identity would permit a more nuanced understanding. Religious identities and beliefs are not static but reflect the temporal and geographic contexts of their adherents.

Regime elites control the mechanisms through which official Islam is disseminated, and can use them to alter its content, but the process, like that of nation-building, takes decades. If regime elites try to change the content of official religious discourse more rapidly, they are likely to experience resistance from the population, who may view the effort as an assault on their religious beliefs. While a national religious identity may be influenced by regime elites during the foundational period, once that identity has been successfully disseminated through mass schooling and standardized religious messaging, it will be more likely to be experienced as reflecting individuals' religious convictions.[17] Religious identity in a population is malleable and subject to change over time, yet if a population suspects that some aspect of their identity is being intentionally altered, they are likely to resist, making top-down change more difficult. Although regime elites can try to shift the content of official religious discourse, it is a lengthy process ill-suited to the short-term demands of politics. Regime elites can try to overcome resistance to a shift in religious discourse by portraying the shift as reflecting local heritage, history, and geography. If the shift can be framed as consistent with existing beliefs, authority is claimed

---

[12]　(Blumber 1969).

[13]　(McDonnell et al. 2017).

[14]　(Wedeen 1999).

[15]　(Hechter 2000; Darden and Mylonas 2016).

[16]　(Cesari 2014).

[17]　(Darden and Gryzmala-Busse 2006).

on the basis of authenticity in a manner that echoes other invented traditions common to nation-building strategies.[18]

Internality: A shift in religious messaging directed at an internal population, such as co-religionists and/or members of the domestic population, arouses less suspicion than a shift seen as primarily directed at an external audience, specifically non-co-religionists. Acts of political violence carried out by Muslim individuals have elevated international interest in the content of Islamic discourses and led to international support for MENA governments that claim their religious messaging counteracts intolerance and/or violence. However, state-sponsored religious institutions face the task of balancing international concern with domestic skepticism, as efforts by state religious authorities to alter existing religious discourse may appear to contravene the faith, especially if the alteration is seen as resulting from external pressure.

The Figure 1 provides a visualization of how the content and direction of official Islam influence the resonance of the religious messages the state seeks to convey. "Content" refers to whether the content of official Islam is consistent or inconsistent over time. "Direction" refers to whether official Islam is directed internally at the domestic population or externally at the international community.

| | | Consistent | Highest resonance | Medium resonance |
| --- | --- | --- | --- |
| Content of Official Islam | Inconsistent | Low resonance | Lowest resonance |
| | | **Internally directed** | **Externally directed** |

Direction of Official Islam

**Figure 1.** Resonance as the Result of the Content & Direction of Official Islam.

Contemporary expressions of Islam are more usefully understood through the prism of the modern state and the agendas of regime elites than by seeking to establish a causal relationship between the behavior of contemporary actors and ancient religious texts.[19] While these actors may themselves view their behaviors as based on foundational Islamic scriptures, their interpretations are shaped indelibly by the ways in which the organs of the modern nation-state have articulated Islam.

Relatively little scholarship has focused on evaluating frame resonance, and even less has tried to focus on state actors as the frame-makers. Arguably no state possesses the tools to fully reshape the religious beliefs of their inhabitants: religion, and belief in general, have consistently demonstrated resilience in the face of the often-ham-fisted efforts by states to assert control.[20]

## 3. Methods and Case Selection

This article evaluates the ways in which the institutions and individuals associated with official Islam respond to the discourses that political elites entrust them with spreading to the population at large. The data are derived from interviews with individual involved in, or targeted by, the messaging. Using the language of official Islam indicates resonance, even if sometimes the effect is not what the government intended. Overall, the article argues that, contrary to prevalent assumptions, official Islam does not uniformly lack

---

[18]  (Comaroff and Comaroff 2009; Hobsbawm and Ranger 1983).

[19]  (Mamdani 2002; Mouline 2014; Hallaq 2013).

[20]  (Migdal 1988; Scott 1998).

credibility. Instead, the resonance of official Islam varies as a result of the consistency and direction of messaging.

Religious bureaucrats, or the individuals charged with disseminating official religious discourse, constitute a core area of focus. Highlighting these individuals, the middlemen and women of the state's discursive sales pitch, offers an in-depth look at how state employees interact with and view the discourses they are charged with promoting. As religious messaging is also disseminated through educational institutions, the data also includes interviews with educators and college students.[21]

Concentrating on these populations reveals how regime elites' desired messages may be transformed in the process of transmission. The article offers a new perspective by focusing on how state-led rhetoric is viewed by those disseminating it, in addition to its targets. Data collected during field research documents the production of official Islam and illustrates the extent to which state-produced religious discourse is either accepted or challenged.

To evaluate whether or not a given interlocutor viewed official religious discourse as resonating with their existing religious identity and ideas, all interviewees were asked to describe official Islam, to explain whether or not they felt that official Islam reflected their own views and, if not, to describe the inconsistencies between their views and those present in official religious discourse. If an individual articulated official religious discourse, and agreed that it matched their views, and used their own words to articulate how official religious discourse reflected their religious identity, the individual exhibited resonance. If the individual challenged official religious discourse, they were coded as demonstrating a lack of resonance.

While excellent recent scholarship has documented the expansion of state control over the religious sphere in the post 9/11 context, much of it analyzes one or two cases without necessarily articulating a broader comparative approach to official Islam in the region. The comparison of three different national contexts constitutes one of this article's primary contributions.

The case selection was limited to monarchies because the monarchic ruling project is based in part on continuity with the past, rather than the legitimating narrative of revolution that characterizes republican governments.[22] Arab monarchies were selected that base their legitimacy, at least in part, on claims of religious authority: Oman, Jordan, and Morocco.[23] Saudi Arabia is also in the category, but due to space constraints, evidence from Saudi Arabia is not included in this analysis.

The monarchs of Jordan and Morocco both claim direct descent from the Prophet Muhammad, making their religious narratives more easily comparable.[24] To demonstrate that the argument can travel within the universe of cases, I selected a third monarchy: Oman, another context where the state encourages a discourse of religious toleration. Oman exhibits different characteristics, including a higher level of wealth, an indirect form of colonization, and a ruling dynasty that does not claim religious authority. Evaluating the applicability of the argument in the Omani context demonstrates that it is not limited to the circumstances of Jordan and Morocco alone.

Ibāḍī rule does not require that the political ruler, or imam, be descended from the Prophet (as in Shi'a Islam), nor of the Quraysh tribe (as is traditional in Sunni Islam), and so Omani leaders have faced minimal pressure to claim descent from the Prophet.[25] Political authority in Oman was traditionally the result of powerful families selecting one of their own members to serve as imam, a position that required the acquiescence of the ruling

21　(Doumato and Starrett 2007; Starrett 1998).

22　(Anderson 1991).

23　(Antoun 2006; Burke 2014; Gutkowski 2015; Maddy-Weitzman and Zisenwine 2013; Valeri 2009; Miller 2013; Peterson 2004; Shryock 1997; Wiktorowicz 1999).

24　(Engelcke 2019).

25　(Hoffman 2012).

families and which could be lost.[26] In Oman, the ruling dynasty overthrew the Ibāḍī imam that once controlled much of the territory, and the sultan took on certain aspects of religious authority such as convening congregational prayer, which according to Ibāḍī theology should only occur in the presence of a ruling imam.

Because the study compares three different countries, no single country is analyzed with the level of detail that focusing on one country alone would permit; instead, insights are derived from comparison.[27] By observing similarities in the ways in which these three different monarchical systems produce religious discourse, the research systematically identifies areas of policy overlap. Variation in the level of observed resonance constitutes the study's primary explanandum. Comparative analysis also allows for certain alternative explanations to be ruled out.

Although the nature of each monarch's claim to religious authority differs, in all three cases the ruling regime justifies its rule at least partially on adherence to Islam, making their form of official Islam broadly comparable. The data was collected during fieldwork conducted between 2015 and 2016.

## 4. Results and Discussion

Figure 2 offers a visualization of how the three cases correspond to the theoretical format established in Figure 1. Note that discussion of the case of Saudi Arabia is not included in this article.

**Figure 2.** Evaluating Resonance in Three Case Studies & a Shadow Case.

### 4.1. Oman

Official Islam in Oman exhibited consistency. When Sultan Qaboos took the throne in 1970, the population of Oman was largely illiterate and had not been exposed to public state-led education.[28] The Omani nation-building project has stressed a heritage of toleration since approximately 1970, due to the potential for sectarian discord between the Ibāḍī and Sunni Muslim sects that constitute most of the population. In addition, Qaboos came to power during the Dhofar War, a secessionist conflict which Qaboos successfully ended and subsequently worked to integrate the Dhofari population into the Omani nation-state.[29] Therefore, this form of religious nation-building has exhibited consistency throughout its duration, although this also reflects the level of absolute control exerted by Sultan Qaboos over official messaging.

Official Islam in Oman exhibited internality. Oman's strategy has focused largely on targeting its domestic population in order to reduce internal religious conflict. Even in the post 9/11 context, the state has focused on domestic messaging rather than actively pursuing efforts to bolster its international reputation for promoting tolerance. This is

---

[26]   (Eickelman 1985).

[27]   (Bennett and Elman 2006).

[28]   (Limbert 2010).

[29]   (Allen and Rigsbee 2000).

largely because Oman's official religious establishment is Ibāḍī, which would not be seen as a credible source of religious leadership by other Muslims. As a result of both consistency and internality, the resonance of official Islam in Oman was high.

In contrast to Jordan, where territorial aspirations and shifting territorial and demographic boundaries undermined potential efforts to articulate a coherent national identity, the Omani monarchy is similar to the Moroccan in prioritizing both the articulation and dissemination of a robust national identity grounded in religious and economic factors. However, while the Moroccan narrative is based on the religious authority of the Moroccan king as the commander of the faithful and is grounded in the relatively mainstream Maliki Sunni school of Islamic jurisprudence, Oman's form of official Islam is Ibāḍīsm. Ibāḍīsm is a sect of Islam which is neither Sunni nor Shi'a but characterized by strict adherence to the Qur'an as well as tolerance toward non-Ibāḍīs. Oman is the only country in the world where the political elites and the head of state are Ibāḍī Muslims. While the Omani state enforces strict non-sectarianism, Ibāḍīsm is sometimes misunderstood abroad as being similar to Shi'ism. This is particularly inaccurate, as Ibāḍī Islam is closer to a strict form of Sunnism, while also maintaining some key distinctions on points of doctrine.[30]

### 4.1.1. Analyzing Statements by Religious Bureaucrats

The effects of nation-building were evident in the responses given during interviews with members of Oman's religious establishment. Sultan Qaboos had successfully equated the nation-state with himself, taking credit for Oman's economic development and stability. Expressions of gratitude for his leadership and wishes for his long life and health were uniform. Although in a system characterized by the centralization of power in one individual it was difficult to gauge the sincerity of such expressions, and these differed from the views of other Gulf citizens towards their equally powerful leaders. Sultan Qaboos enjoyed unparalleled legitimacy and support, because it was under his rule that Oman transformed from a country that lacked paved roads, schools, and hospitals to a middle-income economy, whose GDP per capita is on par with Portugal.

Nation-building typically emphasizes the shared history and identity of formerly distinct territories. However due to the fact that both the interior region and the southern area of Dhofar engaged in armed rebellion against Muscat, efforts to convey unity were especially pronounced in Oman. When asked to explain the Omani stance on religious toleration, the Assistant Grand Mufti explained that Oman's status as a historical entity was significant. He contrasted Oman with polities that were the result of a colonizer's pen stroke, stressing that Oman's religious and mercantile heritage had created a unified identity on the basis of toleration.[31] Processes of nation-building appear to have successfully cultivated a shared sense of nationhood that emphasizes a religious heritage of toleration; however, nation-building has indelibly shaped the ways in which religious heritage is understood, simplifying and codifying it in the typical manner of states. The next sections focus on the effect of nation-building on religious institutions.

Interestingly, views of religious bureaucrats did not vary by location: interviews with employees at the Ministry of 'Awqāf in Muscat expressed nearly identical views as imams and religious educators in small towns in the interior, such as Ibri and Bahla. When asked about Islam in Oman, all religious officials stated that Oman is uniquely tolerant of different religions, and that divisions within Islam do not matter in Oman. However, most acknowledged that Oman was vulnerable to the prevalence of sectarianism outside its borders. An imam in Nizwa, the former capital of the imamate, explained that young people are exposed to more media and may start to believe incorrect ideas. An imam in Ibri explained that Omanis had been peaceful even before embracing Islam, but that Islam had reinforced this tendency. He cited a frequently quoted hadith, in which a follower of the Prophet Muhammad is beaten for proselytizing. Muhammad responded that if the

---

[30] (Hoffman 2012; Mu'ammar 2007).

[31] *Author interview with Assistant Grand Mufti Sheikh Kahlan al-Kharusi, Interview ID20C.* (Muscat, Oman, 11 a.m., 18 November 2015).

man had been in Oman, they would not have beaten him.[32] This hadith is seen as evidence that the Omanis were already known to be a peaceful people.

Representatives of official Islam tended to espouse official religious discourse regardless of their location. Even in Nizwa, the capital of the former imamate, individuals adhered to the official position. An Ibāḍī imam who ran a bookshop near Nizwa Fort, the bastion of the former imam, affirmed that in Oman, Ibāḍī and Sunni and Shi'a individuals would all pray together in the same mosque.[33] He acknowledged that Saudi Islam has had some effect in fomenting sectarianism in Oman, but that the Omani government has actively worked to prevent this. He attributed Oman's ability in resisting Saudi influence to Oman's religious heritage and the leadership of Sultan Qaboos, a view expressed by many. In general, respondents did not go into greater detail than this, which might have required acknowledgement that attributing Omani peacefulness to the sultan's policies was distinct from attributing Omani peacefulness to religious heritage.

While many individuals gave the standardized responses, some provided variations on the overarching narrative of Omani toleration. An imam in Ibri explained that one reason the Omani people peacefully accepted Islam was because at the time of the Prophet Muhammad many Omanis were Christian and explained that Jesus had foretold a prophet named Ahmed would come after him.[34] The willingness of the Omani imam to acknowledge that the period before Islam, usually characterized in many Islamic histories as one of polytheism and ignorance (*jahiliyya*), was notable. His stance was that which the priest interviewed in Jordan had expressed a desire for Jordanian Muslims to acknowledge that the acceptance of Islam was smoothed by a legacy of monotheism.

Of the three monarchies, Omani representatives of official Islam adhered most closely to official religious discourse, as did Omani individuals in general. This may reflect the level of authoritarianism in Oman, and the pervasiveness of state control. Oil wealth and Oman's relatively small population allowed Sultan Qaboos to uniformly disseminate his preferred religious and national narrative during the crucial period of achieving literacy. Omanis express agreement with the official position partly as a result of the Sultan's high level of legitimacy, an asset that his eventual successor will struggle to retain.

### 4.1.2. Analyzing Statements by Educators and Students

When interviewing college students and recent graduates about religious education, they expressed different views regarding the amount and content of Islamic content. One student explained that in his opinion, 45 min for religious education each day was sufficient. However, he acknowledged that religious education was important, because young people relied on the internet for information on religious questions, and in order to use it effectively, young people needed to be able to distinguish credibility. Yet he considered 45 min a day to provide enough time to learn this.[35] Another student said that she felt concerned that schoolchildren were receiving less religious education than they previously had, and she remembered experiencing this reduction after 9/11. She explained that some of the lessons from the Qur'an had been removed even from the time when she was in school.[36] She also acknowledged relying heavily on the internet for answers to questions about religion, but also asking her parents and grandparents.

Interestingly, her description of religious content appeared consistent with what critics of Jordanian religious education said they wanted: for moral values to be expressed in universal terms, rather than simply reflecting Islam. Yet the student found this to be disconcerting, that schoolchildren were not learning some of the Qur'anic stories that she herself had been taught and considered valuable. Instead, children were taught that the values these stories reinforced were universal. Although the young woman voiced

---

32    *Author interview with imam, Interview ID 16B* (Noor Majan Institute, Ibri, Oman, 9 a.m., 13 November 2015).

33    *Author interview with imam, Interview ID 18B* (His bookstore near Nizwa Fort, Nizwa, Oman, 10 a.m., 17 November 2015).

34    *Author interview with imam, Interview ID 16B* (Noor Majan Institute, Ibri, Oman, 9 a.m., 13 November 2015).

35    *Author interview with male college student, Interview ID 7B* (Al Rudha "The Lounge," Muscat, Oman, 12:30 p.m., 4 November 2015).

36    *Author interview with female recent college graduate, Interview ID 5B* (Al Rudha "The Lounge," Muscat, Oman, 2 p.m., 31 October 2015).

concerns regarding this shift to emphasize shared humanity rather than simply Islam, her responses appeared to indicate that such shifts can be effective, because she acknowledged that reading religious texts, whether the Qur'an or the Bible, could bring peace. When asked to explain, she reiterated that all religions were based on the same idea, that they were supposed to bring peace.

Other students expressed frustration with the notion of moderate Islam because they saw it as sending the message that something was wrong with Islam. Two students who asked to be interviewed together, explained that this was why they did not agree with the notion of "moderate Islam." In conversation, it became clear that they saw promoting moderate Islam as something that other countries did, whereas the actions of the Omani government were intended to "preserve" Islam, rather than promote another form of it.[37] This reinforced that the Omani government had been successful in convincing citizens that official religious discourse lacked a political agenda and was only intended to sustain existing religious heritage and identity.

Interviewees expressed a sense of imagined community: An Omani student shared the following: "It was always Oman, there is no specific time period for Oman . . . It was still Oman, though not united, with rulers in each part of the country. But it was understood that even though it was decentralized it was always Oman."[38]

A teacher in Bahla asserted that the lack of politicization of religion in Oman was due to the absence of political parties, specifically Islamist political parties that elsewhere in the region had used religion to serve their own agendas.[39] He also reiterated a key component of official discourse: that Ibāḍī Muslims constitute a majority of Omani citizens. The government does not collect information about sectarian affiliation in the census, so the ratio of Ibāḍī to Sunni Omanis is unknown; however, it is suspected that Ibāḍīs no longer constitute the majority. If the Omani government were to acknowledge this demographic shift, the regime could lose power, or might at least be required to appoint Sunni Muslims to positions of religious authority. However, up to this point most key political figures, and especially religious figures, are Ibāḍī and the government's narrative of moderation and toleration relies on the construction of Ibāḍī Islam as such. The position that Ibāḍīs are the "overwhelming majority" was echoed at all levels of power.

*4.2. Jordan*

Official Islam in Jordan did not exhibit consistency. Prior to 9/11, official religious discourse in Jordan did not emphasize religious toleration or moderation. Likewise, official Islam in Jordan did not exhibit internality. The messaging of religious toleration was primarily directed externally rather than at the Jordanian population. For these reasons, official Islam in Jordan lacked resonance.

4.2.1. Analyzing Statements by Religious Bureaucrats

In Jordan, many religious bureaucrats were willing to criticize the official position, especially regarding the promotion of moderate Islam. An imam in Salt, a town in western Jordan near the border with Israel, had a reputation for being independent from the government and was therefore respected. However, he still operated with the permission of government agencies. The imam articulated the view that religion and the state should be one, meaning that the state should adhere to Islam and be subordinate to Islam, and expressed frustration that often Islam was subordinate to the state, as many imams used religion to justify the actions of the state. The imam asserted that the state only implemented parts of Islam, those related to marriage and divorce and inheritance, but did not in fact adhere to the most important parts of the religion. His view that the state incompletely implemented religion was a frequent theme in Jordan, where both religious

---

37　*Author interview with male and female recent college graduates, Interview ID 9B & 10B* (Al Rudha "The Lounge," Muscat, Oman, 2:30 p.m., 4 November 2015).

38　*Author interview with student, Interview ID 4D* (Muscat, Oman, 16 November 2007).

39　*Author interview with teacher, Interview ID 12B* (His home, Bahla, Oman, 1:30 p.m., 7 November 2015).

officials, educators, and university students expressed the need for more thorough religious education in order for the population to better understand what Islam required.

The imam expressed confidence that technology could overcome the state's use of religion to serve its agenda, because technology allowed people to circumnavigate the limits imposed by state religious institutions. At the same time, he explained that technology allowed anyone to issue a fatwa, joking that even his wife could issue a fatwa on the internet.[40] In articulating the need for better religious education in order to ensure that people were able to differentiate a sound hadith from a false or unsubstantiated one, his concerns echoed those of many societies grappling with the overabundance of false information online, and the difficulty of verifying sources for accuracy.

The imam in Salt expressed more independent language than many of the other government registered imams, although even his views demonstrated the influence of official religious discourse. When asked to explain the difference between "centrist Islam" (Islam al-wasati), a term that appears in the Qur'an, and "moderate Islam" (Islam al-muatadl), a term associated with English language discourse, he asserted that the two were identical. When asked for his view on state Islam, he stated that ordinary people did not take it seriously, because the government did not adhere to Islamic principles. For this reason, he did not expect that the government's efforts to encourage moderate Islam would be successful. This view was markedly distinct from individuals working within the Ministry of 'Awqāf, who stated that because "true Islam" was itself moderate, government efforts to encourage it would be met with success.

An imam based at a mosque near the University of Jordan in Amman expressed skepticism of official Islam. He and his family were still permitted to live in the house attached to the mosque, but he had temporarily lost permission to lead prayers and was required to pay a fine of 20 Jordanian dinars (about $30 USD), due to his espousal of views that did not adhere to the official government position.[41] He explained that speaking about politics in the sermon was forbidden. He also said that while the government did not prevent any religious practices, it did not encourage individuals to study Islam deeply. In his view, the government was controlling religion for the purpose of preventing extremism, a goal that he saw as legitimate.

When speaking about his own role in relation to the government, he distinguished himself from a representative of the Ministry of 'Awqāf. Despite having been trained and accredited by the Ministry, he articulated a clear distinction between individuals that were seen as independent and those that were seen as government-affiliated and asserted that the views of someone independent like himself would be taken more seriously than the views of a government affiliate. He also acknowledged that Jordanian religious institutions lacked credibility: "The Fatwa Department sometimes produces authentic and trustworthy fatwa, and sometimes it produces political fatwa. People trust the authentic ones. It is always apparent if a fatwa (was issued to) support the government's opinion. The Fatwa Department has never issued a fatwa that contradicts the government." [42] Although the imam saw himself as independent from the regime, he also used language consistent with official religious discourse. For example, when explaining that the government closely regulated mosques following the rise of ISIS, he explained that this was positive because "Islam must be true Islam. and that extremist interpretations were incorrect.

An imam in Irbid, a city in northern Jordan near the Syrian border, agreed to an interview on a Friday. After giving the Friday sermon, the imam explained that he had been an employee of the Ministry of 'Awqāf for the past 24 years. Similar to the imam barred from preaching in Amman, the imam expressed that he was opposed to Jordan's treaty with Israel; however, he had managed to avoid censure by not overstepping such

---

[40] *Author interview with imam, his home, Interview ID 41A* (Salt, Jordan, 8 p.m., 5 September 2015).

[41] The disciplinary actions he experienced were consistent with those described in the 2016 International Religious Freedom Report for Jordan, *Bureau of Democracy, Human Rights, and Labor: United States Department of State*. https://www.state.gov/documents/organization/269142.pdf (accessed on 11 November 2017).

[42] *Author interview with imam, his home, Interview ID 7A* (Amman, Jordan, 3 p.m., 30 July 2015).

limits while preaching. He did express dissatisfaction with the government, explaining that "in the Muslim countries we do not respect the government because of the corruption."[43] When asked if people saw imams as having religious authority, he said that many people come to ask him questions about religion, especially pertaining to family affairs. His son, a young man also in the room, interjected to say that people look up answers online. His father did not correct him but reiterated that members of his community come to him.

Even individuals that worked with the government to produce or disseminate official religious discourse expressed doubts as to the long-term sustainability of the regime's wish to monopolize religious discourse. In their view, preventing preachers from commenting on issues that were important to society weakened the credibility of official religious authorities. The director of a state-sponsored organization charged with reproducing the government's official religious stance expressed frustration with the fact that a preacher could lose his position for criticizing the government. The most concerning aspect of this, in his view, was that official religious institutions lost their strongest thinkers, those individuals that could potentially have wielded influence over their congregations. Instead, the government prioritized individuals willing to mouth the official view and ignore the topical issues of greatest importance to the congregation. The government was primarily interested in hiring individuals that were the least likely to inspire loyal adherents to their interpretation of religion. In his view, the government faced a dilemma: religion cannot be ignored, but it is too powerful to be effectively manipulated. Referring to Islam as the "essential engine" of society, the director stated that the government appeared to be trying to weaken religion while simultaneously using it strategically. But he said that they would be unable to do so, that the attempted manipulation of religion would backfire.[44]

### 4.2.2. Analyzing Statements by Educators and Students

This section presents reactions from educators and students regarding their views on official Islam, whether taught in the classroom or conveyed through other channels. In general, the government narrative of "true" Islam was evident, even among individuals who did not otherwise agree with the government's policies. However, college students were less likely to use the words and concepts provided by the government, indicating potentially less resonance of, or less exposure to, the official line.

The head of the "All Jordan" Youth Commission in Ma'an, the restive city in southern Jordan, recounted his experience with extremism among local youth. He acknowledged that many young people had gone to fight in Syria, and that some of them had not come back. He cited high levels of poverty and unemployment as contributing factors in young men leaving Ma'an to join the Islamic State. In his view, the reasons were typically less about religion than about a search for opportunity. However, he also acknowledged the role of religious discourse by explaining that young people tended to think that the strictest interpretations of Islam were correct. For this reason, they tended not to look to the Dā'ira al-Iftā' for religious guidance, for example, but instead to seek the opinions of conservative sheikhs online.[45] As a result of receiving religious advice from some of these conservative sheikhs, some young people came to believe that it was their duty as Muslims to join the Islamic State and fight in Syria.

Saudi Arabia was often mentioned as the lode star for Islamic guidance, highlighting the fact that official Islam in Jordan lacked the credibility to be taken seriously. In an interview, a security expert affiliated with the University of Jordan stated that whatever religious authority the Hashemites once had, they had lost to Saudi Arabia: "It does not matter that the king is Hashemite. We would trust a fatwā from Saudi Arabia more than one from Jordan ... We would trust a fatwā from al-Azhar before one from Jordan."[46] Many respondents cited Saudi Arabia as a religious model.

---

[43]  *Author interview with imam, his friend's home, Interview ID 34A* (Irbid, Jordan, 2 p.m., 28 August 2015).

[44]  *Author interview with director of a state-sponsored religious organization, Interview ID 6A* (Amman, Jordan, 11 a.m., 30 July 2015).

[45]  *Author interview with the director of the All Jordan Youth Commission, Interview ID 55A* (His office, Ma'an, Jordan, 2:30 p.m., 21 September 2015).

[46]  *Author interview with security expert at University of Jordan, Interview ID 29A* (Amman, Jordan, 11 a.m., 24 August 2015).

In an interview with a college student, she explained that the meaning of moderate Islam was not clear. She recalled hearing a preacher talking to his congregation about being open-minded and tolerant, but then in another week he attacked Shi'a Muslims, saying they had strayed from the true path of Islam. In her view, the government was promoting moderate Islam in order to benefit itself, but there were certain aspects of what seemed to be moderate Islam that the government was not particularly interested in encouraging, such as toleration of different interpretations of Islam.[47]

Data collection included a focus group with college students. The group consisted of eight bachelor's students and one doctoral student, four women and five men. The students discussed the Amman Message, and specifically why it was not well-known among Jordanians. One student explained that only people who had taken the tawjīhī since the Message's publication in 2005 would be likely know the Message.[48] Another said that the Amman Message was not really needed, as Jordanians already knew everything that it contained.[49]

When discussing the importance of religious education, the doctoral student explained that, because of satellite television and the internet, people could ask for answers to their religious questions directly, but sometimes this could lead to chaos. In her view, official religious edicts from the government could help to overcome the cacophony of different interpretations, such as designating the day when Ramadan would begin. She acknowledged that some people would disagree with the government, but that most would say that listening to the government was better than chaos.[50] The students expressed different views, at times disagreeing with each other about what was permitted or not, such as whether smoking or music were forbidden by Islam. They agreed, however, that if someone did something wrong, they did not need to be told, because they could feel it internally.

In general, the focus group with the students provided data that did not overly adhere to the government narrative. Their statements did not include any mention of "true" or "correct" Islam. They acknowledged that the government was involved in producing religious discourse but explained that they tended to seek religious information from the Qur'an, hadith, or from the internet.

Despite the view held by some loyalists that the Hashemites had a unique religious standing, other members of the religious bureaucracy did not consider the monarch to possess religious authority. Although many respondents had positive views of the king's efforts to promote the "true" image of Islam as peaceful and moderate, King Abdullah II himself was not seen as a religious leader. Tying Jordanian Islam to the Hashemites would be problematic, given their foreign origins. Proclaiming a form of Jordanian Islam unique to Jordan would exclude the Hashemites, simultaneously undermining their political authority. Without a doctrinal message characteristic of Jordan, Jordanians sometimes look for religious guidance elsewhere, especially to Saudi Arabia.

A possible alternative explanation for the lack of resonance in Jordan could emphasize the relative recency of Jordan's colonial construction. The territory that became Jordan was carved out by European colonizers and lacked any historical center of religious learning or authority, therefore the Jordanian state had to build up an identity as well as religious institutions from nothing within the previous century.[51] This fact could perhaps account for the lack of resonance, in contrast to Oman and Morocco, which were established polities long before the modern era.

Yet the case of Saudi Arabia, although not discussed in the article, represents another example of a state that did not exist prior to the 20th century, yet managed to achieve a robust set of religious institutions and a distinct religious identity relatively quickly.

---

47    *Author interview with college student, Interview ID 31A* (Cafe, Amman, Jordan, 2 p.m., 25 August 2015).
48    *Author interview with college student, Interview ID 47A* (Focus group, Miami Center, Amman, Jordan, 4 p.m., 17 September 2015).
49    *Author interview with college student, Interview ID 50A* (Focus group, Miami Center, Amman, Jordan, 4 p.m., 17 September 2015).
50    *Author interview with doctoral student, Interview ID 45A* (Focus group, Miami Center, Amman, Jordan, 4 p.m., 17 September 2015).
51    (Lynch 1999, 2002).

Although Saudi Arabia possessed the two holiest cities in Islam, the Al-Saud's preferred form of Islam, Wahhabism, was not prevalent in Mecca or Medina, and had to be established as the basis for Saudi Islam, while alternative interpretations were suppressed. The Al-Saud began this process before the massive discoveries of oil wealth that began to transform Saudi society in the 1960s. Arguably the absence of a pre-modern set of religious institutions was not such a hindrance that Saudi Arabia could not redefine its citizens' understanding of Islam. With their oil wealth, they then transformed Islam around the world. Jordan, therefore, could have also established a more robust religious identity based on the Hashemites' religious legitimacy, if they had begun to do so earlier.

*4.3. Morocco*

Official Islam in Morocco is consistent with earlier messaging. The Moroccan monarchy constructed a religious discourse grounded in national heritage, which has remained relatively unchanged since the foundational period which followed independence from France in 1956. Tying the official religious identity to the national identity contributes to the monarch's claims to religious authority, specifically his status as the "Commander of the Faithful", which was codified in the 1962 constitution.

The Moroccan state directs messages of toleration at its domestic population, but also strenuously seeks to cultivate a reputation for moderation abroad. Externally directed messaging can undermine the credibility of official religious discourse, and the resonance of official Islam in Morocco was neither high nor low.

4.3.1. Analyzing Statements by Religious Bureaucrats

In interviews with Moroccan religious bureaucrats, many expressed views that exhibited high resonance, or agreement, with official religious discourse. Official discourse asserted that Moroccan Islam was grounded in Maliki Sunni Islam as well as Sufi mysticism. The third influence cited was the Ash'ari tradition, which was framed as corresponding to rationalism but historically represented a compromise between rationalists and textualists. The religious history of Morocco was brought up in several interviews, specifically the lack of Ottoman influence as contributing to Morocco's unique form of Islam. Notably, although Oman also largely escaped colonization by the Ottomans, this has not been adopted as a key component of Oman's religious heritage in the way that it has in Morocco. The emphasis on Morocco's escape from Ottoman control, and its lack of emphasis in Oman, demonstrates that similar historical factors can be used to suit the regime's narrative as needed.

Official Islam in Morocco highlighted the need to encourage moderation in order to protect "spiritual security" in Morocco. This phrase was a frequent refrain among religious bureaucrats, expressed in Arabic language interviews as "al-'amn al-rūḥiyy" and in French language interviews as "sécurité spirituelle." The repetition of the phrase indicated the centrality of the concept in the official narrative, as well as its apparent resonance.

The efforts by the Moroccan government to discourage extremism included programs such as training for both Moroccan and foreign imams, the education of female religious actors or mourchidates, anti-radicalization interventions in prisons, and efforts to reform education. Such efforts were broadly acknowledged and admired in statements by Moroccan religious bureaucrats. In general, representatives of official Islam expressed explicit agreement with and admiration for the general strategy of the Moroccan king to counter violent extremism. They tended to view such efforts as expanding Moroccan prestige as a religious authority in the region and the world, and in counteracting the dangerous influence of "eastern Islam", which in their use referred to both Egyptian and Saudi Islam.

The Moroccan monarchy appeared to have successfully indoctrinated representatives of official Islam in its official discourse of Moroccan moderate Islam. Clerics and other religious bureaucrats expressed that their task of encouraging moderate Islam was important for both Moroccan prestige and safety. In contrast to many statements by religious officials in Jordan that stressed the necessity of adhering to "true" Islam, religious officials in Morocco tended not to use this phrase. While the notion of true Islam does appear in

Moroccan discourse, it was mentioned with far greater frequency in interviews with Jordanian officials. In general, religious officials in Morocco explicitly discussed the importance of encouraging a form of Islam that could undermine extremism, rather than stressing the need to prioritize "true" Islam. Jordanian religious officials periodically expressed reservations or even subtle criticisms of the regime's efforts to encourage moderation, seeing this as a capitulation to US pressure for example. In contrast, Moroccan religious officials expressed support for the regime's official narrative, describing it as necessary as well as repeating the standard line that it reflected Moroccan heritage.

Jordanian and Moroccan representatives of official Islam differed in their expressed views on state control of mosques. In Jordan, representatives of official Islam expressed some frustration over strict state control of mosques. In Morocco, the official discourse highlighted the importance of "preserving spiritual security."[52] In this context, preserving spiritual security referred to preserving the peace and tranquility of sacred spaces and preventing them from being adulterated by temporal and political concerns. The official discourse skillfully used existing Islamic values regarding the preservation of mosque sanctity in support of the regime's agenda of preventing political speech and agitation in mosques and other religious spaces. In contrast, Jordanian officials pointed out that by suppressing discussion of politics, state control of mosques pushed individuals to seek religious engagement with temporal matters online, for example, rather than in the mosque, often exposing them to more radical online messaging than would be aired in public.

One explanation for why Morocco was more successful in encouraging the adoption of moderate Islam was that policies focused on outreach to communities, such as *the mourchidates* initiative, as well as outreach to prisons and slums. When explaining why Morocco succeeded in encouraging a discourse of moderate Islam, the head of the Mohammedia League of Scholars emphasized the importance of tangible benefits for individuals: "Give deliverables . . . let it touch their children, their souls, let them see and hear and experience . . . Let them feel pride in what you're doing."[53] By stressing an outcome-based approach rather than one primarily focused on making statements, the Moroccan government demonstrated to the population that espousing moderate Islam could provide tangible benefits.

In addition to this materialist explanation, building on the existing parameters of religious identity established during nation-building enhanced the credibility of discourses of moderation. The significance of Morocco's religious identity was frequently highlighted by representatives of official Islam. For example, the Assistant Director of the Dar al-Hadith al-Hassania stressed how a religious national identity contributed to independence from outside influences: "It is not possible to affect us with ideas from Iran or Saudi Arabia, because all Moroccans feel that they have a Moroccan school of religious understanding, and so they cannot be influenced by others."[54] Countering the effect of Saudi Arabia constituted a core objective of Moroccan official Islam. In addition to enforcing the Warsh 'an 'Naafi style of Qur'anic recitation, for example, the Moroccan government does not automatically follow Saudi Arabia's declarations pertaining to the religious calendar. In 2017 Saudi Arabia announced that the new moon marking the end of Ramadan and the beginning of Eid al-Fitr had been spotted on Sunday, 25 June. However, the governments of Morocco, Iran, Oman, South Africa, and Brunei announced that Eid in these countries would instead begin on Monday, 26 June.[55]

Morocco's independence from Saudi religious authority was evident in other ways. The Director of Islamic Affairs at the Ministry of 'Awqāf noted that fewer Moroccans traveled to Saudi Arabia for religious instruction. In his view, this contributed to the spread of more moderate views in mosques and in mass media.[56] The efforts to establish

52 *Author interview with head of Cultural Activities, Directorate of Islamic Affairs, Ministry of Awqāf, Interview ID 4C (Rabat, Morocco 10:15 a.m., 22 July 2016).*

53 *Author interview with the Secretary General of the Mohammedia League of Religious Scholars, Interview ID 11C (Rabat, Morocco 11 a.m., 29 July 2016).*

54 *Author interview with the Assistant Director of Dar al-Hadith al-Hassania, Interview ID 7C (Rabat, Morocco 1:30 p.m., 25 July 2016).*

55 http://www.aljazeera.com/news/2017/06/eid-al-fitr-saudi-arabia-declares-sunday-day-170622071041784.html (accessed on 7 November 2017).

56 *Author interview with Director of Islamic Affairs, Ministry of Awqāf, Interview ID 8C (Rabat, Morocco 11:30 a.m., 26 July 2016).*

Morocco as independent from Saudi religious influence was a component of Mohammed VI's campaign to consolidate the state, with himself at its head, as Morocco's premiere religious authority.

In general, Moroccan representatives of official Islam expressed more willingness to use religion strategically to preserve peace and stability. This contrasted with representatives of official Islam in Jordan, who instead stressed the importance of adhering to "true" Islam, regardless of the potential consequences. The views of both groups of individuals exhibited the influence of the religious discourse they were charged with promoting, but in Jordan the promotion of true Islam actually served to undermine the regime's position, due to individuals' discomfort with policies that they saw as contravening Islam, whereas in Morocco the government's willingness to openly use religion for strategic purposes was embraced as a necessary means of preserving stability while also enhancing Morocco's international status as a source of religious authority.

### 4.3.2. Analyzing Statements by Educators and Students

Many conversations with Moroccan educators tended to reflect the official line. An education expert, for example, explicitly distinguished the Moroccan government from King Mohammed VI, despite the king's control over the government. In her opinion, the king was a neutral arbiter in religious matters, whereas members of parliament and other government officials used religion for their own purposes.[57] This sentiment was reinforced by an employee at the Mohammedia League of Religious Scholars (Rabita Mohammedia lil-'Ulama'), who was involved in educational messaging. When explaining the work of the League, she said, "The League is independent, it reports directly to the palace,"[58] reinforcing the official notion that the king was removed from political matters but indicating a different form of royal dependency.

A few individuals explained that not everyone agreed with the official narrative, although they tended to describe this attitude as held by others. An education consultant said that although he thought the idea of Moroccan Islam accurately reflected the country's unique religious heritage, some people were opposed to the idea of a local interpretation of Islam, because in their view Islam should be the same everywhere. He also pointed out that if people felt that a specific interpretation of Islam was promoted for political purposes, they would be likely to resist it. However, he also explained that many Moroccans were open to the idea of Moroccan Islam: "With the general public there's been a yearning to become reconciled with Morocco's spiritual heritage after decades of Wahhabi and jihad ideas being spread."[59] He explained that a former Minister of 'Awqāf had allowed the influence of Saudi Islam to spread into Morocco, by allowing Moroccan imams to study in Saudi Arabia, for example. He said that some people believed that the government had allowed Wahhabi influence in order to counteract Leftists, a pattern that occurred elsewhere in the region.

When speaking with Moroccan students about the topic of religious education, some conveyed doubts about the approach. During a round table with nine students, a male student from Rabat explained that in 2016 the government had removed certain verses from the religious curriculum, such as those that spoke about jihad. He saw this as unwise, because students would find the verses themselves online. If students could have the verses explained to them in the classroom rather than by some sheikh online, they would be more likely to interpret them in a manner that would not radicalize them. He also criticized the decision to reduce the number of hours devoted to religious education on similar grounds, arguing that students would seek their own answers and might then come under the sway of extremists.[60] The student recognized that different interpretations of Islam existed, and that the government could use a version that worked for its purposes as

---

[57] *Author interview with education expert, Interview ID25C* (Home of her friend, Rabat, Morocco, 8 p.m., 10 August 2016).

[58] *Author interview with employee of the Mohammedia League of Religious Scholars, Interview ID 10C* (Rabat, Morocco 11 a.m., 27 July 2016).

[59] *Author interview with educational consultant, Interview ID24C* (Cafe, Rabat, Morocco, 6pm, 8 August 2016).

[60] *Author interview with student, Interview ID12C* (Language Café meeting, Café 7eme, Rabat, Morocco, 6:30 p.m., 3 August 2016).

easily as the jihadis could. This awareness contrasted with the view expressed by many Jordanians, namely that there was one "correct" or "true" form of Islam. The prevalence of this view in Jordan was due to its centrality in official religious discourse, and an indication that the Jordanian government had in fact influenced the population's views, but in a way that undermined the government's religious authority.

During the focus group, students were asked whether they remembered being taught about Moroccan Islam as being Maliki, 'Ash'ari, and Sufi. They answered negatively: that the emphasis had been on Moroccan Islam as Maliki.[61] This corresponded to the fact that the adoption of rhetoric pertaining to 'Ash'ari Islam occurred in 2014, after the students were already in college and no longer taking religious education classes. The students explained that many of the religious lessons focused on citizenship at the same time as Islam, almost equating the two, and that the equality of the genders was a component of the lessons on citizenship. In general, the students felt that they should have received more religious instruction, or that Moroccan students in general should receive more. This view was due to their conviction that acts of violence were committed by individuals with insufficient understanding of Islam. However, the students also admitted that sometimes individuals who most fervently expressed religious beliefs were also those most likely to act in ways that contravened religion.[62] They saw this as a uniquely Moroccan form of hypocrisy, although did not explain why they did not think this would occur outside of Morocco.

## 5. Conclusions

The article uses the concept of resonance, defined as the extent to which an individual experiences official discourse as corresponding to what they recognize as familiar and generally accepted within a given social setting. Trying to evaluate resonance can sometimes suffer from tautological explanations, as the most resonant discourses are successful because they are . . . the most resonant. Evaluating contexts in which the resonance of official religious discourse varies offers a means of escaping circuitous logic, by comparing the key causal factors at work. A most-similar systems approach reduced the possibility that this variation was the result of other possible causes.

Resonance adheres at the level of individual experience; therefore, interview data provides evidence of resonance. The research is a work of theory building; future application of the theory could test the argument through national level surveys, although the collection of massive amounts of data would lack the interpretivist edge provided by face-to-face interaction. Still, testing the theory with a much larger N would allow for the establishment of a "threshold of resonance" among the population more broadly, rather than focusing on individuals within the religious and educational establishment as bellwethers of resonance.

By focusing on interviewees' willingness to repeat official religious discourse, the study accounts for potential reluctance to discuss topics as sensitive as religion, the state's preferred rhetoric, and the authority of rulers. Religious bureaucrats might reproduce official discourses regardless of their own private views. However, religious bureaucrats are also aware of the potential blow to their own credibility if they parrot discourses that do not correspond to the views of their co-nationals.[63]

In Jordan, resonance was relatively low. Religious bureaucrats stressed the importance of the Hashemites in promoting "true" Islam, while non-bureaucrats highlighted the need to adhere to "true" Islam but did not necessarily consider the monarchy as upholding it. In contrast, the language of "true" Islam was far less common in Morocco or Oman; the pervasiveness of the concept of "true" Islam in Jordan appears to reflect its use in official

---

[61]  *Author interview with student, Interview ID14C* (Language Café meeting, Café 7eme, Rabat, Morocco, 6:30 p.m., 3 August 2016).

[62]  *Author interview with student, Interview ID17C* (Language Café meeting, Café 7eme, Rabat, Morocco, 6:30 p.m., 3 August 2016).

[63]  (Nielsen 2017).

religious discourse, and even if in practice critics of the regime use the concept as a means of pointing out the ways in which the Hashemites do not represent "true" Islam.

The higher resonance of official religious discourse in Morocco and Oman is due to these regimes' successful effort to build on the religious and national identity established during the foundational period, while the relatively lower resonance of official religious discourse in Jordan is related to the lack of successful nation-building, and specifically the Jordanian regime's lack of investment in promoting a specifically Jordanian form of Islam. Instead, the Hashemite regime has asserted that its historical religious authority grants it the legitimacy and responsibility to speak for all Islam, rather than articulating a specifically Jordanian form of religious expression.

The findings pertaining to resonance also relate to evidence of violent extremism in each case. Oman has been most successful in cultivating a narrative that appears to prevent violent extremism. Although extremism cannot be causally reduced to the content of official religious discourse, it is telling that no acts of extremist violence have been carried out by Omanis, either domestically or abroad. Morocco has successfully cultivated an international reputation for moderation, to the extent that other countries send imams to study Moroccan Islam. However, Morocco's claims are somewhat undermined by acts of terrorist violence carried out by Moroccans, which belie claims that Morocco has inculcated toleration as a core value of its national religious identity. Jordan enjoyed a first mover advantage in promoting moderate Islam, beginning in 2004 with the Amman Message.[64] Even so, the disconnect between national identity and religion contributes to the inability of the monarchy to convincingly assert its religious authority.

Prevailing wisdom assumes that state-led projects of religious social engineering would be unsuccessful, due in part to the nature of religion itself. Yet by analyzing the ways in which states can and cannot successfully transform their citizens' views on religion, the article offers insights for both the power over and the limits of state control in people's lives and beliefs.

**Funding:** Research in Jordan was funded in part by a Foreign Language and Area Studies (FLAS) scholarship administered by the Institute for Middle East Studies at the George Washington University's Elliott School for International Affairs, CFDA number 84.015B.

**Institutional Review Board Statement:** The study was conducted according to the guidelines of the Declaration of Helsinki, and approved by the Office of Human Research—Institutional Review Board at George Washington University, study number 021466, approved 21 May 2015.

**Informed Consent Statement:** Informed consent was obtained from all subjects involved in the study.

**Data Availability Statement:** The data will be made available following the publication of the author's book.

**Conflicts of Interest:** The author declares no conflict of interest.

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
