# Peer review of "Evaluating the Resonance of Official Islam in Oman, Jordan, and Morocco"

_religions, doi:10.3390/rel12030145_

Round 1
Reviewer 1 Report
I enjoyed reading this fascinating essay.
The central argument here is predicated upon a controlled comparison of religious discourse in Oman, Morocco, and Jordan. At stake is whether “official” Islam as espoused by state organs of religious authority (typically the Awqaf or Fatwa ministries) resonates with individuals’ preconceived “frames” that link conceptions of identity (spiritual, national, cultural, and the like) with their vision of what constitute appropriate doctrinal content and practices within religion. This makes it easy to score the cases based upon the author’s primary variables of consistency and direction. When official Islam is highly consistent and is internally directed, it resonates (i.e., Oman); when it is consistent but externally directed, it resonates less (i.e., Morocco); but when it is not consistent and is externally directed, it hardly resonates at all (i.e., Jordan). These covariations are, I think, accurate and compelling, particularly since there are structural controls built into the comparative exercise (all are pro-Western monarchies in the Middle East, which espoused “moderate” Islam after 9/11).
So, the overall thesis is persuasive. Moreover, I think the qualitative evidence is rich, in terms of showing how consistency and direction shape the resonance of individuals that the author has interviewed, particularly religious bureaucrats, educators, students, and other citizens. The sharply notices when these voices are willing to repeat the official line about moderate Islam and its doctrinal guidance, versus when they jeer at it. These are critical twists within uttered discourse (and certainly, discursive performativity), and this helps the evidence serve as a reliable indicator about the resonance of state-directed Islamic discourse with the assumptive beliefs or prescriptive framework of religion held within society.
My only quibble is at the margin. I think the author should clarify what causes low resonance (a big part of the thesis!), which in empirical terms implicates the case of Jordan. Jordan is in effect, the outlier. Why do so many Jordanian imams and laypersons jeer at the state’s efforts to advance al-islam al-mu’tadil? The technical answer draws upon low consistency and externally directed pressures, but I notice this answer shifts over the essay. The author in the earlier parts of the article emphasizes the nature of directionality, and mentions the weakness of nation-building in Jordan; but in the conclusion, the author doubles down on the failure of nation-building, and the associated lack of religious legitimacy accorded to the Hashemite monarchy (that is, low consistency).
There is a very important claim to tease out here. Nation-building is a social, institutional, and coalitional process that unfolds over generations: so, bluntly, the Jordanian state’s efforts to promote and reconstruct moderate Islam after 9/11 was a hollow exercise not just because it came from outside interests, but because it was being uttered by a set of ruling-making institutions that had no spiritual credibility. Lack of resonance is therefore a *long-term* outcome of deep historical processes, in an almost deterministic way. Compare this to the argument about internal/external direction, which is more of a *short-term* consideration regarding how a domestic audience perceives new shifts in state-directed religion. So, one takeaway could be that success with a long-term process/variable (nation-building), in effect, guarantees a regime some minimal degree of resonance later on, no matter whether it comes internally or externally. That makes me want to conclude that nation-building/consistency is, by and large, the real explanation for how state-led Islamic content fits within the frames of its adherents, because spiritual legitimacy cannot be won or lost in a day; it stems from antecedent political interactions, social conflicts, and regime consolidations that predate whatever immediate interests take precedence today. What makes the Jordanian case even more stunning is that this territory was especially ripe for religious dominance by a state claiming Sharifian legitimacy early on (and yet it stumbled). It had no ‘ulama class and no centers of Islamic spirituality that predated either the British or Ottoman imperialism; yet the failure of the Hashemite monarchy to create a coherent national identity that secured religious and political authority simultaneously long ago doomed the success of any state-led religious turn after 9/11.
So, clarifying the potentially deterministic nature of the argument (which I think it is, and there is nothing wrong with that!) would be the primary revision I recommend.
Author Response
Dear Anonymous Reviewer,
Many thanks for your thorough and thoughtful review of my submission.
I very much agree with your suggestion that I must contend with the fact that Jordan was a colonial construction and so therefore I must demonstrate that the lack of resonance of official Islam was in fact the result of the causal factors I identify and does not simply reflect the lack of a sufficiently august and robust set of religious institutions and traditions upon which to build.
I especially liked your phrasing: “because spiritual legitimacy cannot be won or lost in a day; it stems from antecedent political interactions, social conflicts, and regime consolidations that predate whatever immediate interests take precedence today.”
To address this, I added a short discussion of this possible alternative explanation, starting at line 526. I use the case of Saudi Arabia as another relatively recent state, which yet managed to establish a clear religious identity:
A possible alternative explanation for the lack of resonance in Jordan could emphasize the relative recency of Jordan’s colonial construction. The territory that became Jordan was carved out by European colonizers and lacked any historical center of religious learning or authority, therefore the Jordanian state had to build up an identity as well as religious institutions from nothing within the previous century. This fact could perhaps account for the lack of resonance, in contrast to Oman and Morocco, which were established polities long before the modern era.
Yet the case of Saudi Arabia, although not discussed in the article, represents another example of a state that did not exist prior to the 20th century, yet managed to achieve a robust set of religious institutions and a distinct religious identity relatively quickly. Although Saudi Arabia possessed the two holiest cities in Islam, the Al-Saud’s preferred form of Islam, Wahhabism, was not prevalent in Mecca or Medina, and had to be established as the basis for Saudi Islam, while alternative interpretations were suppressed. The Al-Saud began this process before the massive discoveries of oil wealth that began to transform Saudi society in the 1960s. Arguably the absence of a pre-modern set of religious institutions was not such a hindrance that Saudi Arabia could not redefine its citizens’ understanding of Islam. With their oil wealth, they then transformed Islam around the world. Jordan, therefore, could have also established a more robust religious identity based on the Hashemites religious legitimacy, if they had begun to do so earlier.
I fear this may be a somewhat inelegant response to your very valid critique, but it seemed to be the best way to address your point within the space provided. In the book length version of the research project, I do discuss Saudi Arabia, yet did not have space to introduce it as a full case here.
I hope this addition at least addresses your concern, although I agree that a longer discussion would be merited.
Thank you again for your time and well-reasoned response!
Best regards.

Reviewer 2 Report
Overall, the paper is well written. Yet, there are numerous conceptual, theoretical, and methodological issues with this manuscript:
Concepts:
- What is official Islam? How do you conceptualize it? Is it state-sponsored Islam?
- Why official Islam lacks credibility on politics? It is not a sort of an essentialist thinking and way too overgeneralized understanding of Islam’s relationship with politics? Islam and politics is an evolving phenomenon and cannot (should not be) dichotomized. Why would official Islam help in any way curb radicalization (as in some cases it acts as a catalyst of discontent)? For instance, official Islam in Tajikistan is way different than in Egypt (right?)
- The concept of “resonance” is poorly developed. Theoretical linkages of resonance is underdeveloped.
Theory:
- The theoretical framework is poorly constructed. The author does not sufficiently explain why and how official Islam would have to resonate to begin with? Can you name one single case (in the West or East) where it resonated fully? To what extent would that help prevent violence? What if government is pushing official propaganda driven narratives constantly? Do we still want that propaganda to resonate with actors in civil society (or non-state religious networks)?
- The paragraph, lines 44-19, is debatable and lacks empirical rigor. The literature review on resonance is poor.
- Overall, the theoretical underpinnings are very shallow.
Methods:
- Original field interviews are great. They yield highly insightful observations. However, given the research question (and objective), the data at hand is insufficient to support such a big claim advanced in the manuscript. The explanatory strength of data is insufficient in this case. Many other confounding variables are not accounted for (how about other variables that influence “resonance”).
- A comparative study between Morocco and Jordan can be justified (most similar system design approach). However, the Sultanate of Oman may not be relevant case here. There are many confounding factors when it comes to Omani monarchy which diverges significantly from Morocco and Jordan. Case selection justification for Oman is poorly framed.
Author Response
Dear Anonymous Reviewer,
My sincere thanks for your thorough and thoughtful review of my submission.
I suspect that one of the underlying reasons for several of the concerns you raised is that this article is an attempt to distill a book-length project. It seems I sacrificed the depth and clarity that I hope characterize the longer work while condensing it into article form. However, I do hope that by addressing your valid critiques I will be able to achieve greater depth while preserving relative brevity.
I have tried to address each of your concerns, and so include them below in italics, separated into the categories you used. At time, it seems you may have reiterated a critique, in which case I make reference to an earlier revision.
Concepts:
- What is official Islam? How do you conceptualize it? Is it state-sponsored Islam?
To answer your first question, I define official Islam in the opening sentence as “state-sanctioned religious messaging.” I also added a paragraph conceptualizing the term in greater detail, beginning at line 34.
- Why official Islam lacks credibility on politics? It is not a sort of an essentialist thinking and way too overgeneralized understanding of Islam’s relationship with politics? Islam and politics is an evolving phenomenon and cannot (should not be) dichotomized.
In response to your question about why official Islam lacks credibility, as I explain this is a widespread assumption in existing literature, which I agree needs to be problematized: this is part of my intention in writing this article and pursuing the longer research project. I added the following in order to explain more clearly “…based on the assumption that these non-democratic governments suffer from a general legitimacy deficit, which extends to their production of official Islam.” (line 46)
- Why would official Islam help in any way curb radicalization (as in some cases it acts as a catalyst of discontent)? For instance, official Islam in Tajikistan is way different than in Egypt (right?)
In response to your question about why official Islam would curb radicalization, the general assumption has been that official Islam suffers from a total lack of credibility, yet my findings indicate that this assumption needs to be investigated more thoroughly. As my findings indicate, there is in fact variation in the extent to which official Islam is seen as credible. Unsurprisingly, the governments of the cases I considered have not gathered empirical data to evaluate whether or not their efforts to discourage violent extremism are successful. And as you suggest, such a characterization would be overly simplistic. Therefore, I do not try to answer whether or not the production of so-called “moderate Islam” prevents acts of violence, which are themselves the results of so many complex factors. Instead, I focus on the ways in which religious bureaucrats, teachers, and other figures articulate the state’s official religious narrative, whether they parrot it, or rearticulate it, or reject it. Although you expressed concern that the scope of the project is too vast for an article, I do not in fact attempt to take on the question of the causality of violent extremism, which I agree would be too much for a single article. Instead, my focus is merely on how these individuals engage with this discourse they are charged with disseminating.
I very much agree with your point that official Islam in Tajikistan differs from official Islam in Egypt. This key point motivates my article, in which I try to show that different countries possess a very different set of religious narratives on which to build. If I had the space in this article, as well as the research funding to compare the content of official Islam across all the countries that define Islam as the official religion of the state, I would have loved to do so. However, given my qualitative approach and limits of both time, funding, and article length, I can only compare these three cases.
- The concept of “resonance” is poorly developed. Theoretical linkages of resonance is underdeveloped.
In response to your critique that I poorly develop the concept of resonance, I added three additional paragraphs to develop the concept more robustly, which include five additional footnotes, (lines 116-143).
Theory
- The theoretical framework is poorly constructed. The author does not sufficiently explain why and how official Islam would have to resonate to begin with?
In response to your critique that the theoretical framework is poorly constructed, I believe this may again be the result of trying to convert a much longer project into a shorter one. As mentioned above, I added additional discussion and citations to make the discussion of resonance more robust.
- Can you name one single case (in the West or East) where it resonated fully? To what extent would that help prevent violence? What if government is pushing official propaganda driven narratives constantly?
You ask me to “name a single case where official Islam has resonated fully?” I posit that the case of Oman represents the closest example of where official Islam has resonated fully. Although I am sure that not every individual fully agrees with how the Omani state articulates Islam, in my experience, respondents in Oman do express strong agreement with the government’s conception of official Islam. Having spent half a year in Oman, including four months living with a host family and interacting closely with Omanis, I feel fairly confident in my assessment, despite its limitations, that many Omanis generally feel that the official rhetoric of the state regarding religion reflects their own perceptions. As I argue, this is partly tied to the fact that the Omani government has not tried to develop a reputation for moderate Islam outside Oman, and instead has focused on directing messages at Omanis, who therefore have little cause to suspect that the religious messaging is primarily intended to improve the standing of the Omani regime in the eyes of outsiders, especially the US.
- Do we still want that propaganda to resonate with actors in civil society (or non-state religious networks)?
Your question, “Do we still want that propaganda to resonate with actors in civil society (or non-state religious networks)?” lies outside the scope of the article. The value judgement of whether or not “we want” a given outcome seems more normative than the article’s intent. I do not make any calls pertaining to what I “want” as a scholar, I merely evaluate the responses of my interlocutors and develop a framework to explain the patterns of variation I observe. For that reason, I cannot come up with a satisfactory response to this question, as my “wanting” anything regarding civil society or any other group has very little to do with my analysis. As I state at the outset, a robust literature has focused on non-state religious networks; given the attention that non-state religious networks have already received, this article argues for the need to consider the effects of state religious messaging.
- The paragraph, lines 44-19, is debatable and lacks empirical rigor.
The paragraph that previously began at line 44 and in the current version of the manuscript begins at line 60, I explain that within the scope of the defined universe of cases, that is “in the Arabic-speaking countries of the Middle East and North Africa (MENA) where Islam is the official religion,“ (line 36-7) the state has in fact heavily influenced Islam, in that the state, especially in the period since 9/11, increasingly controls mosques, dictates the content of religious discourse in schools, and controls religious messaging available in media. This is not to say by any means that the state’s control of religion is absolute – it certainly is not, especially for the many millions of individuals in the MENA region with internet access – however it is prevalent. To back up this point, I have added citations to several works devoted to the prevalence of state religious discourse in these contexts.
To respond directly to your critique that “The paragraph, lines 44-19, is debatable and lacks empirical rigor,” I revised by being more precise in my choice of words. I also added citations to five of the existing works that informed my conception of the influence of the state upon the religious sphere, especially but not limited to resulting from the expansion of state control over religious spaces, actors, and institutions following elevated concerns about religious extremism since 2001, (especially Feuer and Wainscott) as well as scholarship on the expansion of state control over religious institutions as a result of the process of building modern states in the MENA region (see especially Cesari, Fabbe, and Zubaida).
- The literature review on resonance is poor.
In response to your reiterating the critique that, “The literature review on resonance is poor.”, I added a more robust conceptualization of the concept, including additional citations, as discussed above, starting at line 116.
- Overall, the theoretical underpinnings are very shallow.
In response to your reiterating the critique that, “Overall, the theoretical underpinnings are very shallow,” I added additional citations as footnotes to help support certain points, especially regarding the legacies of nation-building as establishing a shared set of ideas, (Darden & Gryzmala-Busse). I more thoroughly articulated the theoretical underpinnings of the “consistency” variable, drawing on Benford & Snow, Blumber, McDonnell, Bail, and Tavory to more clearly develop the conceptualization of resonance (116-143). I also added a paragraph to acknowledge that states are typically not very adept at controlling or replicating complex human practices, citing the work of Joel Migdal as well as James Scott (line 202).
Methods:
- Original field interviews are great. They yield highly insightful observations. However, given the research question (and objective), the data at hand is insufficient to support such a big claim advanced in the manuscript. The explanatory strength of data is insufficient in this case. Many other confounding variables are not accounted for (how about other variables that influence “resonance”).
I believe this critique is tied to your question about whether resonance of official Islam would prevent violence, I reiterate that this is not what I am trying to evaluate. The decision to commit an act of political violence is the result of such a multitude of different factors, some of which are mutually exclusive: some individuals are motivated, in part, by frustration at the absence of economic opportunity, for example, while others come from wealthy backgrounds and are motivated by the opposite, i.e. they possess economic opportunity but feel unfulfilled, etc. Therefore, this article does not attempt to explain whether or not official Islam will prevent violence. This article only purports to investigate the views of the individuals charged with disseminating official Islam, and to evaluate why certain patterns emerge in the likelihood that religious bureaucrats from Oman, for example, were more likely to express agreement, or resonance, with official religion than members of the religious bureaucracy in Jordan. Therefore, your question about resonance and violence lies outside the scope of the article.
- A comparative study between Morocco and Jordan can be justified (most similar system design approach). However, the Sultanate of Oman may not be relevant case here. There are many confounding factors when it comes to Omani monarchy which diverges significantly from Morocco and Jordan. Case selection justification for Oman is poorly framed.
You suggest that I remove the case of Oman. However, this would obviate your question about where official Islam fully (or in the Omani case, mostly) resonates. I consider it important to include Oman, in order to offer a broader spectrum of resonance, than to follow the suggestion of removing the case of Oman. I added a sentence pointing out the importance of exploring the dynamics at work in cases beyond merely Jordan and Morocco (line 249).
In addition, the comparison of three cases constitutes a key contribution of the piece, as well as one of its more original attributes. In fact, I would have included 4 cases, as I do in the book, but since 3 cases already strains the capacity of a single article, 4 seemed indeed too many, especially given my reliance on interview data.
Thank you again for your time and for offering a thorough critique of my article.
I hope that I have addressed your concerns and that by responding to your feedback, I have improved the article significantly.
Best regards.
Round 2
Reviewer 2 Report
Thank you for providing insightful responses to my concerns. After carefully revisiting the paper, I can see manuscript in a much different light. I do think, however, that you need to improve "connecting" your cases studies to your overall findings, particularly with respect to Oman. As you know, Oman is quite unique within Islamic world with its Ibadi form of Islamic practice (I think it is the only nation-state with such denomination). Thank you for your hard work. Well done.
Author Response
Dear Anonymous Reviewer,
Thank you for taking the time to review the submission again. I was pleased to learn that you found the revisions mostly satisfactory.
To address your suggestion that I need to more thoroughly justify and explain the inclusion of the Omani case, I added the following:
Lines 287-297: I explain why I think it is important to evaluate my argument in more than only the relatively similar cases of Jordan and Morocco; that Oman is broadly comparable in its efforts to promote a narrative religious tolerance yet offers interesting variation in that it did not focus on trying to establish an international reputation for these efforts. Therefore, it offers a key source of variation on one of my key variables. I also provided more explanation for establishing authority within the Ibadi tradition and added an additional citation (Eickelman).
Lines 303-308: I explain more generally the study’s reliance on a comparative approach in order to evaluate the argument I make.
Lines 337-348: In the empirical section on Oman, I provide more background about Ibadism, and how Oman’s approach to religion should be understood in contrast to that of Jordan and Morocco. I added an additional footnote, directing readers to Hoffman’s excellent overview of Ibadism, as getting into the specifics of theological differences is outside the article’s scope.
I hope that these revisions address your concerns.
Thank you again for your time and keen attention.
Best regards.